# International Evaluation of China's Beef Cattle Industry Development Level and Lagging Points

**Xujun Li [1], Xiaoping Ma [1,2], Mingli Wang [1,*] and Hao Zhang [1]**

[1] Institute of Agricultural Economics and Development, Chinese Academy of Agricultural Sciences, Beijing 100081, China

[2] School of Economics, Inner Mongolia University of Finance and Economics, Hohhot 010070, China

\* Correspondence: wangmingli@caas.cn; Tel.: +86-136-4120-3038

**Abstract:** Quality, efficiency, safety and environmental protection are important directions for the development of animal husbandry in China. Taking China's beef industry as an example, this study establishes a comprehensive index system from six industrial subsystems: resource endowment, production, consumption, quality, trade and environment. By comparison with the beef cattle industry in other countries, great effort is being made to position the development level of China's beef industry and to determine its lagging points, according to the coupling coordination degree and relative development degree of each subsystem. Under the multidimensional development goals, the development level of China's beef cattle industry shows a fluctuating upward-downward trend, and the resource endowment has a certain advantage, but the development of the production, consumption, quality and environmental subsystems is insufficient and lacks competitiveness. China's beef cattle industry is less developed than in Brazil, the United States, Argentina, Australia and other countries in terms of production, consumption, quality, trade and environment. The industrial subsystems mainly present low-level coordination and operation, with lagging development of the production quantity and quality. According to the analysis of the industry's weaknesses, China's beef industry needs to promote the combination of planting and breeding, cost reduction, efficiency increase, and green breeding.

**Keywords:** beef cattle industry; development level; international evaluation; coupling and coordination; lagging point

## 1. Introduction

In September 2020, the Opinions of the General Office of the State Council of China on Promoting High-Quality Development of Animal Husbandry were issued, and the important concept of "high-quality development" was formally established as the main direction for the development of China's animal husbandry industry. The most prominent feature of the concept of "high-quality development" is the comprehensiveness of the strategic direction [1]. High-quality development means that the development of China's livestock industry should meet the diversified goals of high efficiency, high quality, safety and environmental protection.

From 1978 to 2000, the Chinese government promulgated a series of positive policies and measures, such as the Report on Accelerating the Development of Animal Husbandry and the implementation of the "vegetable basket project", which greatly improved the macro environment for the development of China's livestock industry and led the volume of livestock industry to increase greatly. Over this period, China's beef cattle industry developed rapidly. China's annual beef production increased from 230,000 tons to 5,131,000 tons, cattle stock increased from 71.346 million to 123.532 million, an increase of 73.1, cattle slaughter increased from 2.968 million to 38.069 million, an increase of 11.8 times, respectively, and the possession of beef per capita increased from 0.24 kg to 4.05 kg, an increase

of approximately 16 times. During this period, the development orientation of China's beef cattle industry mainly focused on the improvement in quantity and had not yet fully considered the safety and quality of livestock products. In such a context, problems of substandard quality and safety of livestock products arose [2].

In 2001, China joined the WTO, and the Chinese beef cattle industry began to pay attention to the quality and safety of products on the basis of satisfying domestic demand. Moreover, China actively participated in international trade competition and competed for international market share. In 2004, China promulgated the first national-level plan for livestock industry standards, the "Plan for the Construction of National Standards and Industry Standards for Animal Husbandry (2004–2010)", which provided the industry standardization criteria for various industry chain links, such as production, processing, packaging and the transportation of the beef cattle industry. By 2012, the quality and safety of China's livestock products had comprehensively improved, with the pass rates of "lean meat essence" in products, breeding, feed and feed product quality monitoring being 99.7%, 99.98%, 100% and 95.7%, respectively.

With the rapid development of China's livestock industry, a series of environmental problems have arisen, such as the pollution of the environment by large amounts of livestock and poultry manure and the ecological imbalance caused by greenhouse gas emissions, such as $CO_2$. In response to these problems, the Chinese government has introduced a series of environmental protection measures, and livestock manure management, ecological farming, and environmental protection have become priorities. The livestock industry is the main source of greenhouse gas emissions from agricultural activities in China, and as the livestock species with the largest greenhouse gas emissions, the beef production is the main source of carbon emissions from the livestock industry, with its carbon dioxide equivalent emissions accounting for 41% of the total emissions from the livestock industry. In November 2013, the Regulations on the Prevention and Control of Pollution from Livestock and Poultry Scale Farming was introduced as China's first livestock and poultry environmental policy regulation. A series of plans, opinions and policies for livestock and poultry manure management followed one after another. In January 2015, the Environmental Protection Law of the People's Republic of China was officially implemented, which mentioned that a tax would be imposed on cattle farms with a stocking size greater than 50 head. China's beef cattle industry has started to pay attention to the combination of breeding and green production to ensure the quality of livestock products and protect the ecological environment from the source of production.

The development of China's beef cattle industry has gradually shifted from focusing solely on quantity improvement to focusing on multiple goals of quality, safety and green development. Therefore, what is the level of development of China's beef cattle industry, considering the multiple objectives of high efficiency, quality, safety and environmental protection? How does the competitiveness of the beef cattle industry compare with that in other countries (regions)? Where are the key points that need to be improved? This study starts from the whole beef cattle industry system, according to the world ranking of beef production of each country, and takes into account the availability of the data; 11 countries (or regions) including the United States, Brazil and Argentina are selected to compare with China in terms of their beef industry, whereby these 11 countries (or regions) are distributed in Asia, Europe, America and other continents, They are representative beef cattle breeding countries in the world. We can understand the international level of China's beef cattle industry better by comparison.

The objective of this study is to identify the lagging aspects in the development of China's beef cattle industry, and provide various references for the development of China's and other countries' (or regions) beef cattle industry. To achieve this objective, we establish a comprehensive index system, comprehensively analyze the development level of China's beef cattle industry, and compare China's beef cattle industry with other countries.

## 2. Theoretical Analysis and Research Hypothesis

*2.1. Theoretical Analysis*

The theory of absolute superiority put forward by the British economist Adam Smith in The Wealth of Nations, published in 1776, is regarded as the origin of the theory of the division of labor [3]. The theory of absolute advantage holds that absolute advantage is the basis of international trade, and this absolute advantage is mainly manifested in the gap in labor productivity or the production cost caused by the gap in endowment conditions, capital accumulation, technological innovation levels and so on between countries. The absolute cost of a country's production of a product determines whether the product should be imported by the country or produced and exported by it. If the absolute cost of producing the product is higher than that of other countries, a country should import the product to avoid greater losses; if it is lower than other countries, it should be produced and exported. However, the limitation of this theory is that it cannot explain why countries or regions that do not have absolute advantages in the production cost of any product still actively participate in the international division of labor.

The theory of the comparative advantage was first proposed by David Ricardo, a British economist, in his book On the Principles of Political Economy and Taxation, in 1817 [4]. The theory holds that there are differences in labor productivity among countries, resulting in comparative advantages. A country should produce products with greater advantages or products with smaller disadvantages. As long as there are relative differences in the level of advantages or disadvantages, it can obtain comparative benefits through trade.

In the 1930s, Swedish economists Eli Herkscher and Bertier Olin proposed the factor endowment theory [5]. This theory holds that the comparative advantage of different countries for a certain product is affected by factor endowments. A country should export products with abundant resource endowments and import products with scarce factor endowments.

The theory of competitive advantage was first proposed by the American economist Michael Porter [6]. A country's trade advantage is not simply felt by the country's natural resources, labor force, interest rate, exchange rate and other basic factors, but to a large extent depends on higher-level factors such as the social system, technological innovation and industrial development stamina. With the continuous refinement of the international division of labor, the factors affecting the international competitiveness of industries are increasing. Many scholars have made more supplements to the relevant theories, which are the basis of this study.

*2.2. Research Hypothesis*

According to theoretical analysis, there are many factors that affect the competitiveness of a country's beef cattle industry. At present, China has set a multi-dimensional goal for the development of animal husbandry. According to this goal, six subsystems of resource endowment, production, consumption, quality, trade and environment are constructed to evaluate the development level of China's beef cattle industry. Prior to China's entry into the WTO, the development level of the domestic beef cattle industry was improved under the influence of policy support and economic development. Following China's entry into the WTO, the development level of the beef cattle industry in other advantageous countries (or regions) declined. In 2019, China imported 1.659 million tons of beef, mainly from Brazil, Argentina, and Australia. Therefore, the competitiveness of China's beef industry is lower than those of these countries. China's beef cattle industry started relatively late. Compared with other advantageous countries (or regions), the production efficiency of beef cattle varieties is low, and the output and quality levels are not high. Accordingly, this paper proposes three research hypotheses.

**Hypothesis 1 (H1).** *Under multidimensional development goals, the development level of China's beef cattle industry shows a fluctuating upward-downward trend.*

**Hypothesis 2 (H2).** *China's beef cattle industry is less developed than Brazil, the United States, Argentina, Australia and other countries.*

**Hypothesis 3 (H3).** *The production quantity and quality of China's beef cattle industry is lagging.*

## 3. Materials and Methods

### 3.1. Construction of the Evaluation Index System

To determine the development level of China's beef cattle industry, a set of specific indicators reflecting its development level needs to be designed to scientifically quantify and index such factors as high efficiency, high quality, safety and environmental protection. The core evaluation indicators should be selected, disregarding the indicators with a high correlation or similarity [7] while following the principles of objectivity, operability and data availability. Based on the multiple objectives of China's livestock industry development, the evaluation of China's beef cattle industry development level is carried out from six subsystems: resource endowment, production, consumption, quality, trade, and environment.

#### 3.1.1. Resource Endowment Development Conditions

The development of the industry is constrained by the resource structure. The relative area of farmland and grassland affects the breeding structure; a country (region) with more farmland than grassland is more suitable for the development of grain-based livestock, such as poultry and hogs, and vice versa for the development of grass-fed livestock, such as beef cattle and sheep. The forage supply capacity is an important realistic basis, so the resource structure, the forage planting proportion and the forage self-sufficiency rates are chosen to assess the resource endowment development conditions, and the forage supply situation is assessed on the basis of the existing resource structure to measure the resource advantage situation of a country in the development of the beef cattle industry [8].

#### 3.1.2. Production System Development Conditions

The relative importance of beef in total meat production represents the production potential of the country's beef cattle industry [9], so the cost efficiency represented by the price ratio of meat to feed, the technical efficiency represented by the amount of meat produced per unit of beef cattle and the proportion of the domestic beef production are chosen to assess the development level of the production system of the beef cattle industry.

#### 3.1.3. Development Conditions of the Consumption System

The greater the demand for livestock products is, the greater the supply that can be driven to a certain extent to promote the expansion of the industry, and the faster the increase in demand is, the faster the development of the industry. This study integrates the real and potential consumption capacity, selects the beef consumption per capita and the proportion of domestic beef consumption to measure the real consumption of a country (region), and selects the beef consumption per capita growth rate to reflect the consumption growth potential and consider the potential consumption capacity. The development of the consumption system is also an important aspect that has often been ignored in previous analyses of the development status of the beef cattle industry.

#### 3.1.4. Quality System Development Conditions

In economics, the indicator "quality" is generally classified as a "constant assumption" or follows the assumption of high quality and high price [10]; the higher the price is, the higher the quality of the product, called the "quality-price" symmetry assumption. Based on this assumption, considering the data availability and quantifiability, the livestock product quality upgrading index is selected to represent the quality system development changes in the beef cattle industry according to the research needs, and the product quality

upgrading measure is used to infer the quality changes by observing the market share changes after controlling for relative price changes.

### 3.1.5. Trade System Development Conditions

Based on the dual consideration of the domestic supply capacity and the international market supply capacity, the domestic market self-sufficiency rate, international market share and trade competitiveness index are selected for the measurement and comparison.

### 3.1.6. Environmental System Development Conditions

The degree of the environmental accommodation of livestock and poultry feeding and the intensity of greenhouse gas emissions in a country (or region) are important indicators to measure the development conditions of the environmental system of a country (region) and the sustainable development of the beef cattle industry. The environmental system development conditions of China's beef cattle industry are mainly indicated by the carrying capacity per unit agricultural area for beef cattle breeding and the emissions of greenhouse gases ($CO_2$, $CH_4$, $N_2O$, etc.) generated from the intestinal fermentation and manure of beef cattle for each unit of beef cattle production value created [11].

Finally, the specific indicators selected to evaluate the development level of China's beef cattle industry are shown in Table 1.

**Table 1.** Measurement indices and main descriptions of the development level of China's beef cattle industry.

| Target Level | Criteria Indicators | Specific Measurement Indicators | Indicator Measurement Method | Attributes |
|---|---|---|---|---|
| Development level of China's beef cattle industry (A) | Resource endowment development conditions (B1) | Resource structure (C11) | Grassland area/farmland area | + |
| | | Share of forage grain cultivation (C12) | Forage harvested area/grain harvested area [1] | + |
| | | Self-sufficiency of forage grain for beef cattle (C13) | Forage grain production/forage grain consumption [2] | + |
| | Production system development conditions (B2) | Cost efficiency (C21) | Beef producer price/feed grain producer price [3] | + |
| | | Technical efficiency (C22) | Beef cattle meat production/beef cattle stock, i.e., unit beef cattle meat production | + |
| | | Domestic beef production share (C23) | Domestic beef production/total domestic meat production | + |
| | Consumption system development conditions (B3) | Beef consumption per capita (C31) | Beef consumption/population size | + |
| | | Domestic beef consumption share (C32) | Domestic beef consumption/total meat consumption | + |
| | | Growth rate of beef consumption per capita (C33) | [(Current beef consumption per capita/previous beef consumption per capita)/previous beef consumption per capita] × 100% | + |
| | Quality system development conditions (B4) | Product quality upgrading index (C41) | (beef export value in period t + 1/ beef export volume in period t + 1)/ (beef export value in period t/ beef export volume in period t) | + |

**Table 1.** *Cont.*

| Target Level | Criteria Indicators | Specific Measurement Indicators | Indicator Measurement Method | Attributes |
|---|---|---|---|---|
| Development level of China's beef cattle industry (A) | Trade system development conditions (B5) | Domestic market self-sufficiency rate (C51) | Domestic beef production/(domestic beef production + net beef imports) | + |
| | | International market share (C52) | (Domestic beef exports/total world beef exports) × 100% | + |
| | | Trade competitiveness index (C53) | Net beef imports/total beef exports and imports | + |
| | Environmental system development conditions (B6) | Number of beef cattle accommodated per unit of agricultural land area (C61) | Beef cattle stock/ agricultural land area | + |
| | | Production value of beef cattle created per unit of carbon emissions (C62) | Beef cattle production value/carbon dioxide equivalent of greenhouse gases from the enteric fermentation and manure of beef cattle [4] | + |

Note: "+" in the table indicates positive attributes. [1] Grain crop harvested area is obtained from the FAO as the sum of the harvested area data corresponding to the cereal number (No. 1717), potato (No. 1720), bean (No. 1726), and soybean (No. 236); forage grain harvested areas is the sum of the corn and soybean harvested areas [12]. [2] Feed grain production is mainly corn and soybean production; feed grain consumption = corn production + net corn imports + soybean production + net soybean imports. [3] The feed grain producer price is replaced by the sum of corn and soybean producer prices, which are discounted at a ratio of 7:2 between corn and soybeans according to the feed grain ration. [4] Greenhouse gases include methane, carbon dioxide and nitrous oxide, all calculated using the $CO_2$ equivalent converted in the international standard IPCC.

### 3.2. Methods and Data Sources

3.2.1. Analytic Hierarchy Process

The analytic hierarchy process is one of the classical multi-attribute decision-making methods widely used in academia and has the characteristics of a clear decision evaluation process, simple application, easy operation and high practicality. Referring to the indicator construction method of Pei [13], this study establishes a recursive hierarchical structure model of the competitiveness of China's beef cattle industry and divides the evaluation system into target, criterion and scheme layers: the target layer is the development level of China's beef cattle industry (A); the second layer (criterion layer) is the first-level indicators (B), namely, the development conditions of the resource endowment (B1), the development conditions of the production system (B2), the development conditions of the consumption system (B3), the development conditions of the quality system (B4), the development conditions of the trade system (B5) and the development conditions of the environmental system (B6); the third layer (scheme layer) is a number of second-level indicators (C), as described in Table 1.

For the assignment method of indicator weights, this study uses the coefficient of variation method. This method can clearly distinguish the ability of each indicator to reach the average level, and compared with traditional subjective assignment methods, such as the expert scoring method, this method has the advantages of a strong objectivity, as it avoids the subjective preference of experts, and the enhanced scientificity of the evaluation results [14–18]. To better locate the gap point between the development of China's beef cattle industry and that of other countries (regions), the specific calculation method is

$$CV_i = \sigma_i / \overline{X_i}, \ (i = 1, 2, 3 \ldots n)$$

$$W_i = CV_i / \sum_{i=1}^{n} CV_i$$

where $CV_i$ is the coefficient of $\sigma_i$ variation, $\overline{X_i}$ are the standard deviation and mean of $i$ indicators, respectively, $W_i$ and are the weights of $i$ indicators.

When this study compares the development level of China's beef cattle industry internationally, it is equivalent to establishing a common "frontier" to compare the development level of the beef cattle industry in different countries, so the same weight is given to each country's indicators, and this weight is objectively and scientifically assigned according to the differences of each country's indicators. For example, when the International Institute for Management Development [19] compares the international competitiveness of the world's top 59 economies, its indicator weights are designed assuming that the weights are the same for each country. The Human Development Index [20] compiled by the United Nations Development Program and the Good Life Index [21] compiled by the Organization for Economic Cooperation assign the same weight to each country when comparing their indices.

Considering the inconsistent units and differences in the variability of each layer as well as each graded index, the direct comparison of them is not possible, so the standardization of each index is needed. The min-max standardization method is used to put each indicator on a scale level.

If the indicator is a positive indicator, the treatment is

$$X = X_{ij} - X_{i,\min} / X_{i,\max} - X_{i,\min}$$

If the indicator is a negative indicator, the treatment is

$$X = X_{i,\max} - X_{i,j} / X_{i,\max} - X_{i,\min}$$

$X$ are the results after standardization, $X_{ij}$ are the original data values of the indicators, and $X_{i,\max}$ and $X_{i,\min}$ are the maximum and minimum values of the $j$ indicators.

Then, the linear weighting method is used to obtain the indices of different target layers of the beef cattle industry, and the calculation process is

$$I = \sum_1^i I_{Bi} W_{Bi}, \ I_{Bi} = \sum_1^j I_{Cj} W_{Cj}$$

$I$ is the development level of the beef cattle industry, $I_{Bi}$ and $W_{Bi}$ are the primary indicators and weights, and $I_{Cj}$ and $W_{Cj}$ are the secondary indicators and weights, respectively.

### 3.2.2. Coupling Coordination Degree Model

Coupling is often used to reflect the dynamic relationship between two or more systems in the consistent development and good interaction to achieve mutual support, mutual coordination and mutual promotion. The coupling system constructed in this paper consists of six intra-industry subsystems of resources, production, consumption, quality, trade and environment and is extended on the basis of the application of the system coupling coordination degree by Wei and Qi [22] and the modification of the model by Wang et al. [23] to obtain the coupling coefficient C of the evaluation index of each subsystem in this study, with the following model form:

$$C = \left[ \frac{U_{natural\ resources_t} \times U_{produce_t} \times U_{consume_t} \times U_{quality_t} \times U_{trade_t} \times U_{environment_t}}{\left( \frac{U_{natural\ resources_t} + U_{produce_t} + U_{consume_t} + U_{quality_t} + U_{trade_t} + U_{environment_t}}{n} \right)^n} \right]^{\frac{1}{n}} \quad (1)$$

In Equation (1), $U$ is the development index of each subsystem, and the subscripts are the resource endowment (natural resources), production (produce), consumption (consume), quality (quality), trade (trade), and environment (environment). The value of $C$ is between [0, 1]; when $C \leq 0.3$, it indicates a low-level coupling; when $0.3 < C \leq 0.5$, it indicates a contradictory state; when $0.5 < C \leq 0.8$, it indicates an improving state; when $0.8 < C \leq 1$, it indicates a high-level coupling. To avoid the disadvantage of a high overall coupling degree exhibited when the development levels among the multiple systems are

high or low at the same time and to better reflect the systemic coordination among the indicators, a more scientific and rigorous coupling coordination degree model is introduced to further measure the degree of synergy among the development levels of each subsystem, and the coupling coordination coefficients $D$ for the resource endowment, production, consumption, quality and environment are constructed as follows:

$$D = \sqrt{C \times T}, T = \alpha\, U_{natural\ resources} + \beta U_{produce} + \gamma U_{consume} \\ + \eta U_{quality} + \phi U_{trade} + \lambda U_{environment} \tag{2}$$

where $\alpha$, $\beta$, $\gamma$, $\eta$, $\phi$, $\lambda$ are the important weight indicators indicating the resource endowment, production, consumption, quality, trade and environment, respectively, based on the aforementioned determination of the weights of the development level of the respective subsystems, which take the following values:

$$\alpha = 0.2840,\ \beta = 0.1367,\ \gamma = 0.1440, \\ \eta = 0.0552,\ \phi = 0.1981,\ \lambda = 0.1821 \tag{3}$$

$$\alpha + \beta + \gamma + \eta + \phi + \lambda = 1$$

At present, there is no uniform definition of the division of the coupling coordination degree in the academic community. This paper draws on the division standard of Zhao et al. [24] to divide the coupling coordination coefficient within the range of $0 \leq D \leq 1$; that is, the value of $D$ ranges between 0 and 1. When $D \leq 0.3$, it indicates a low coordination; when $0.3 < D \leq 0.5$, it indicates a medium coordination; when $0.5 < D \leq 0.8$, it indicates a medium-high coordination; when $0.8 < D \leq 1$, it indicates a high coordination.

The relative development index is used to measure the relative development degree and trend of the development between two subsystems. For the sake of easy expression, the relative development degree between the two subsystems is calculated by using U1, U2, U3, U4, U5 and U6 to represent the combined development level indices of the resource endowment, production, consumption, quality, trade and environment, respectively.

$$\beta = U_i / U_j \tag{4}$$

where $i, j = 1, 2, 3, 4, 5, 6$. Referring to Zhao et al. [25], when $0 < \beta \leq 0.9$, subsystem $i$ lags behind the development of subsystem $j$; when $0.9 < \beta \leq 1.1$, the two develop simultaneously; when $\beta > 1.1$, subsystem $j$ lags behind subsystem $i$.

### 3.2.3. Data Sources

The data sources used in this section are the UN FAO database (FAO) and the UN Comtrade database (UN Comtrade) for the time period 1995–2019. Affected by the availability of data, 1995 is the earliest comprehensive data available for the following countries or regions, and 2019 is the latest comprehensive data available for the following countries or regions. The data sample of 24 years (1995–2019) could reflect the development of the beef industry and support the needs of this study.

According to the world ranking of beef production of each country, taking into account the availability of the data, 12 countries (or regions) are selected: China (Asia), USA (North America), Brazil (South America), Argentina (South America), Australia (Oceania), Spain (Europe), France (Europe), Germany (Europe), Italy (Europe), UK (Europe), Japan (Asia) and Korea (Asia). The missing data for individual years are supplemented by the interpolation method.

## 4. Results and Discussion

### 4.1. Overall Trend of the Development Level of China's Beef Cattle Industry

Once the measurement is completed, the comprehensive index of the development level of China's beef cattle industry in 1995–2019 ranged from 3.0 to 5.0 (Table 2). The overall development level of China's beef cattle industry shows a slowly fluctuating upward-

decreasing trend. Taking 2008 as the main node, from 1995 to 2008, the overall development level index of China's beef cattle industry fluctuated from 3.72 to 4.14, an increase of 11.24%, with the improvement in the development level of the resource endowment and production system pulling the overall industry development level; from 2008 to 2019, it fluctuated from 4.14 to 3.33, a decrease of 19.60%, mainly due to the significant decline in the development level of the trade system, proving that the H1 was valid.

**Table 2.** Changes in the level of development of China's beef cattle industry and the level of development of each subsystem, 1995–2019.

| Indicator Name | 1995 | 2000 | 2005 | 2008 | 2010 | 2015 | 2016 | 2017 | 2018 | 2019 |
|---|---|---|---|---|---|---|---|---|---|---|
| Resource endowment development level (B1) | 6.87 | 7.14 | 7.25 | 7.24 | 7.17 | 7.65 | 7.65 | 7.55 | 7.61 | 7.59 |
| Resource structure (C11) | 1.68 | 1.68 | 1.64 | 1.62 | 1.62 | 1.62 | 1.62 | 1.62 | 1.62 | 1.62 |
| Share of feed grain cultivation (C12) | 2.59 | 2.75 | 3.17 | 3.37 | 3.46 | 3.97 | 3.97 | 3.94 | 3.96 | 3.94 |
| Feed self-sufficiency ratio (C13) | 2.60 | 2.71 | 2.43 | 2.25 | 2.10 | 2.06 | 2.06 | 1.99 | 2.03 | 2.02 |
| Level of development of production systems (B2) | 1.18 | 0.92 | 1.67 | 2.74 | 2.63 | 3.15 | 3.92 | 3.59 | 3.41 | 3.43 |
| Cost efficiency (C21) | 1.18 | 0.25 | 0.72 | 1.52 | 1.07 | 1.51 | 2.28 | 1.83 | 1.66 | 1.55 |
| Technical efficiency (C22) | 0.01 | 0.46 | 0.76 | 1.02 | 1.39 | 1.53 | 1.52 | 1.63 | 1.62 | 1.66 |
| Domestic beef production share (C23) | 0.15 | 0.21 | 0.18 | 0.20 | 0.17 | 0.11 | 0.12 | 0.13 | 0.13 | 0.22 |
| Level of consumer system development (B3) | 1.07 | 1.04 | 1.16 | 1.15 | 1.02 | 1.12 | 1.19 | 1.28 | 1.44 | 1.63 |
| Beef consumption per capita (C31) | 0.02 | 0.16 | 0.19 | 0.22 | 0.22 | 0.24 | 0.24 | 0.26 | 0.29 | 0.34 |
| Domestic beef consumption share (C32) | 0.05 | 0.18 | 0.17 | 0.17 | 0.14 | 0.12 | 0.13 | 0.15 | 0.18 | 0.31 |
| Growth rate of beef consumption per capita (C33) | 1.07 | 0.70 | 0.81 | 0.76 | 0.66 | 0.76 | 0.82 | 0.87 | 0.98 | 0.98 |
| Quality system development level (B4) | 0.31 | 0.35 | 0.33 | 0.34 | 0.34 | 0.31 | 0.35 | 0.33 | 0.39 | 0.32 |
| Product quality upgrading index (C41) | 0.31 | 0.35 | 0.33 | 0.34 | 0.34 | 0.31 | 0.35 | 0.33 | 0.39 | 0.32 |
| Level of trade system development (B5) | 6.71 | 5.93 | 6.34 | 6.44 | 4.88 | 1.55 | 1.49 | 1.35 | 1.23 | 1.07 |
| Domestic market supply rate (C51) | 1.49 | 1.48 | 1.48 | 1.48 | 1.47 | 1.30 | 1.26 | 1.23 | 1.13 | 0.98 |
| International market share (C52) | 0.19 | 0.08 | 0.18 | 0.20 | 0.21 | 0.15 | 0.14 | 0.11 | 0.10 | 0.09 |
| Trade Competitiveness Index (C53) | 5.03 | 4.38 | 4.68 | 4.77 | 3.20 | 0.11 | 0.09 | 0.01 | 0.01 | 0.00 |
| Level of environmental system development (B6) | 0.57 | 0.96 | 1.17 | 1.33 | 1.26 | 1.21 | 1.19 | 1.23 | 1.26 | 1.29 |
| Number of beef cattle accommodated per unit of agricultural land area (C61) | 0.39 | 0.42 | 0.33 | 0.29 | 0.22 | 0.18 | 0.19 | 0.18 | 0.19 | 0.19 |
| Emission intensity (C62) | 0.17 | 0.54 | 0.83 | 1.04 | 1.04 | 1.03 | 1.01 | 1.05 | 1.07 | 1.10 |
| Target layer A Level of the development of China's beef cattle industry | 3.72 | 3.68 | 3.95 | 4.14 | 3.76 | 3.32 | 3.41 | 3.35 | 3.33 | 3.32 |

Note: The index of the development level of China's beef cattle industry in the target layer is the weighted index value of indicators at all levels; due to space limitations, the index of the industry system development in key years is listed in the table.

In terms of the changes in the development level of each subsystem, the resource endowment index (B1) increased from 6.87 in 1995 to 7.59 in 2019. Among them, the resource structure (C11) is more stable, and although the feed self-sufficiency ratio of China's beef cattle industry (C13) has shown a decreasing trend in recent years, the domestic industrial restructuring has led to an increasing trend in the proportion of feed grain cultivation (C12), and the increase exceeds other indicators, which is the main driving factor increasing the overall development level of the resource endowment. The production system development level (B2) grew significantly, from 1.18 to 3.43, an increase of nearly two times, with the cost efficiency (C21), the production technical efficiency (C22) and the proportion of the domestic beef production (C23) all showing an upward trend and the

highest degree of technical efficiency growth, indicating that the technical level of beef cattle production has improved significantly. The development level of the consumption system (B3) increased slightly, in which the per capita beef consumption (C31), the proportion of domestic beef consumption (C32) and the growth rate per capita of beef consumption (C33) all increased, reflecting that the per capita beef consumption level in China has improved significantly as Chinese residents pay more attention to food nutrition and improve their living standards. The level of quality system development (B4) is less variable and more stable overall. The development level of the trade system (B5) presents the most obvious decline in the six subsystems, with 2008 as the node. There is a significant decline in the development level; the index fell from 6.44 to 1.07, a decline of 83.41%, and especially after 2010, there was a precipitous decline. The domestic market self-sufficiency rate (C51), the international market share (C52) and the trade competitiveness index (C53) all declined, and the trade competitiveness index (C53) has scored 0 in recent years. The level of the environmental system development (B6) improved, and the output value created per unit of carbon dioxide equivalent emitted increased slightly, indicating that the Chinese beef cattle industry has a tendency toward developing into a low-carbon model.

*4.2. Comparative Analysis of the Beef Cattle Industry Development Levels between China and Other Countries*

The measured results of the indices of the development level of the beef cattle industry in each representative country are shown in Figure 1. Overall, Brazil has the highest level of cattle industry development (mean value of 8.34), followed by Australia (mean value of 7.67), Argentina (mean value of 7.56) and the United States (mean value of 7.27), while China (mean value of 3.68) is in the lower position, ranking eighth among the 12 representative countries. Specifically, Brazil's beef cattle industry has the highest level of development and is on an upward trend, growing from 6.68 in 1995 to 10.20 in 2019, an increase of 52.72%, mainly due to the rising level of development of the country's resource endowment and trade system pull. The overall development of the beef cattle industry in Australia is at a high level but shows a certain downward trend from 8.67 to 7.87, a decrease of 10.14%, with changes in the resource structure and the degree of development of the consumption system being the main influencing factors. The overall level of development of the beef cattle industry in Argentina is more stable, remaining between 7.0 and 8.0. The development level of the U.S. beef cattle industry is characterized by a decline—rise change and is most obviously influenced by the degree of development of the trade system. The development level of China's beef cattle industry as a whole is low, with a slightly fluctuating trend, and is influenced by the change in the development degree of the trade system. France, Italy and Japan have a slightly higher level of development than China, with a smaller degree of variation, proving that the H2 was valid.

In terms of the development level of the subsystems of the beef cattle industry in each country (Table 3), except for the resource endowment, the development levels of the production, consumption, quality, trade and environmental subsystems in China are all insufficient. Brazil has the highest level of development of the resource endowment system and is on a growing trend, and its resource structure is not much different from that of China, but the forage cultivation area is above 80%, and the forage self-sufficiency rate is more than 1 year-round, this conclusion is consistent with the view of an agricultural professor in Brazil, His view is that Brazil has a high level of integration of agriculture and animal husbandry, and the quantity of beef exported to China is very large [26]. Argentina and the United States are also in the forefront, with their forage planting area of more than 70% and the forage self-sufficiency rate of more than 1. China has a good resource structure, but the proportion of the forage cultivation is approximately 40%, and the forage self-sufficiency rate for beef cattle is below 1. The domestic forage production supply can hardly meet China's beef cattle production demand.

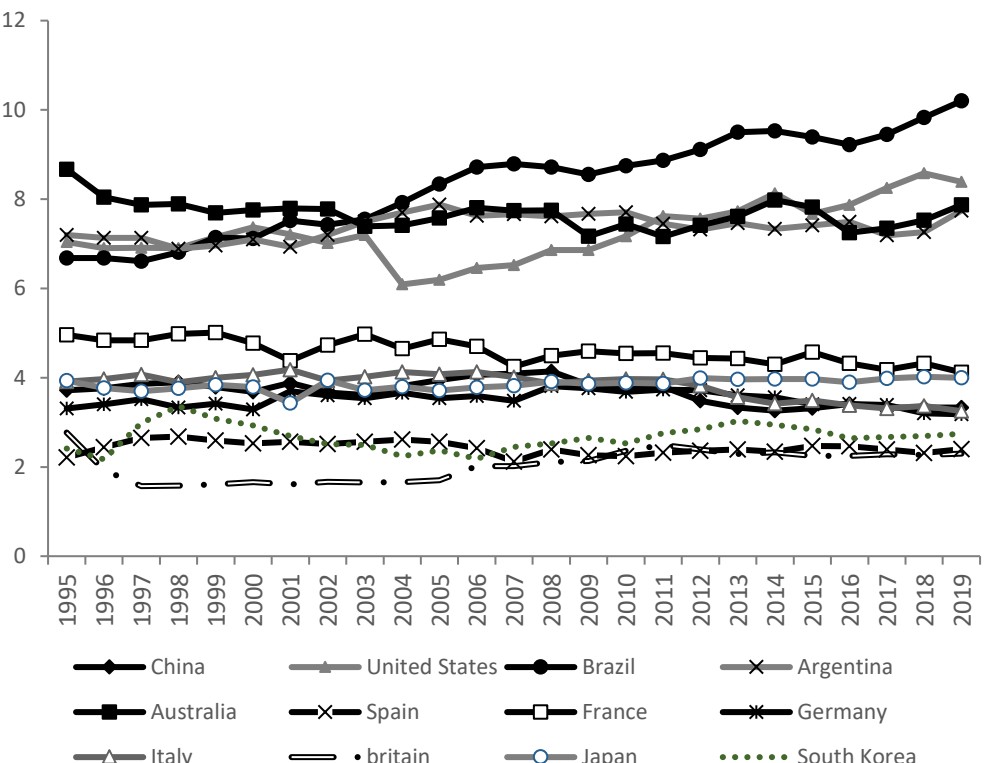

**Figure 1.** Comparison of the level of development of the beef cattle industry in the representative countries from 1995 to 2019.

**Table 3.** Comparison of the development level of the subsystems in the beef cattle industry among the representative countries.

| Subsystem | China | United States | Brazil | Argentina | Australia | Spain | France | Germany | Italy | United Kingdom | Japan | South Korea |
|---|---|---|---|---|---|---|---|---|---|---|---|---|
| Resource endowment (B1) | 7.29 | 10.62 | 12.48 | 12.23 | 11.36 | 1.51 | 6.21 | 1.86 | 4.69 | 0.10 | 0.72 | 0.81 |
| Production system (B2) | 2.24 | 7.17 | 4.44 | 7.00 | 6.25 | 3.71 | 5.50 | 5.11 | 6.42 | 3.68 | 5.23 | 3.37 |
| Consumption system (B3) | 1.18 | 6.94 | 10.80 | 6.60 | 4.92 | 2.77 | 3.41 | 2.86 | 4.30 | 3.33 | 2.36 | 2.76 |
| Quality system (B4) | 0.33 | 0.41 | 0.40 | 0.43 | 0.42 | 0.41 | 0.40 | 0.40 | 0.39 | 0.41 | 0.52 | 0.63 |
| Trade system (B5) | 4.61 | 9.46 | 11.02 | 8.75 | 14.28 | 4.67 | 5.53 | 6.52 | 2.43 | 2.62 | 0.50 | 0.40 |
| Environmental system (B6) | 1.11 | 2.06 | 2.34 | 1.36 | 0.17 | 0.81 | 2.57 | 3.08 | 2.72 | 2.67 | 13.54 | 8.13 |

The development level of China's beef cattle production subsystem is at a distinct disadvantage, with extremely low production technical efficiency and the proportion of the domestic production ranking at the bottom. Compared with other countries, one factor is that the technical efficiency of the Chinese beef cattle production is low, and the meat production per unit of beef cattle is only approximately half that of the United States, while the technical efficiency of the beef cattle production in countries with weaker resource endowment conditions, such as Japan and South Korea, is also approximately two times and 1.5 times that of China, respectively. Second, the proportion of domestic beef production in China is far from that of other representative countries, with only 7.43%, while the domestic proportions of beef production in Argentina, Australia and the United States are above 65%, 56% and 31%, respectively, and those in Japan and South Korea are above 10%.

China's beef consumption system and trade system are both at a low level of development. Brazil, Argentina and the United States have been at the forefront of beef consumption levels, with the per capita beef consumption of the three countries being

17.5, 3.7 and 10.5 times that of China and the shares of the domestic beef consumption being 11.3, 16.2 and 8.2 times that of China, respectively. The beef trade competitiveness indices of Australia and Brazil are close to 1, with extremely competitive export advantages. Compared with them, China is at a disadvantage in both domestic beef self-sufficiency and international market share.

The development level of China's beef cattle product quality system is also low. In terms of the livestock product quality upgrading index, Korea has the highest beef product quality level, i.e., its beef product quality level has an advantage in the international market. With an overall low level, China's beef cattle product quality system ranks 11th, indicating that the current Chinese beef product quality has a significant gap compared to other countries and does not have a competitive advantage.

The development level of the Chinese beef cattle industry environment system is also low. Japan's beef cattle industry ranks first in terms of the environmental system development. On the one hand, due to the limitation of Japan's land area, its beef cattle breeding is mainly produced on a large scale and on an intensive basis, the number of beef cattle accommodated per unit of agricultural land area is higher, and the number of beef cattle accommodated per hectare of agricultural land is approximately 910, which is 5.87 times higher than that of China (155 cattle/ha); on the other hand, its beef cattle production value created per unit of carbon emission (2531.31) is 10.04 times higher than that of China (252.24), which is higher than that of most countries. Korea also has an advantage in the development of environmental systems in the beef cattle industry. China is in 10th place among the representative countries in terms of the environmental system development.

*4.3. Analysis of the Coupling and Coordination Degree of China's Beef Cattle Industry*

The coupling degree and coordination degree of each subsystem of China's beef cattle industry, since 1995, are shown in Table 4. From the trend of the temporal change in the coupling degree, the coupling degree of each subsystem of the Chinese beef cattle industry shows a fluctuating change of rising-declining-rising, and although it is still in the developing stage, it shows a trend of advancing from the low degree to the medium degree of the developing stage. On the whole, the coupling relationship of each subsystem in China's beef cattle industry is mainly in the moderate development state. However, to prevent the phenomenon of overall coupling when the development level of each subsystem is high or low at the same time, the coupling coordination degree and its time-series change characteristics show an inverted U-shaped curve relationship between the subsystems of China's beef cattle industry, i.e., first fluctuating up to the highest point and then declining continuously. On the whole, the coupling coordination degree of each subsystem of the Chinese beef cattle industry shows an inverted U-shaped curve. On the whole, the subsystems of China's beef cattle industry are in a state of low coordination, i.e., the development rate of each subsystem is inconsistent, and the degree of coordination is low.

*4.4. Comparison of the Coupling Coordination Degree of China's Beef Industry and Other Countries' Beef Industry*

Brazil, Australia, Argentina, the United States and France, the top-ranking countries in the development level of the beef cattle industry, were selected to measure their coupling coordination indices and compare with China, and the results are shown in Figure 2. The coupling and coordination index of Brazil's beef cattle industry is the highest and shows an increasing trend, indicating that the development speed and trend of the six subsystems of Brazil's beef cattle industry are relatively consistent, and each subsystem is advancing towards the trend of mutual coordination in development, which may be one of the reasons for the overall competitiveness of its beef cattle industry. Brazil is followed by Argentina and the United States, in which the overall coordination degree of Argentina is stable, and the coordination degree of the United States is on an increasing trend and has surpassed Argentina in recent years. The coordination degree of Australia and France is close, and

the trend is more stable. The coordination degree of China's beef cattle industry subsystem is lower than that of the remaining five countries, and the coordination degree is at a low level.

**Table 4.** Coupling coordination state of the subsystems of China's beef cattle industry, 1995–2019.

| Year | Resource Endowment | Production | Consumption | Trade | Quality | Environment | Degree of Coupling (C) | Coupling Coordination Degree (D) | Coupling Coordination Level | Coupling Stage |
|------|------|------|------|------|------|------|------|------|------|------|
| 1995 | 0.069 | 0.012 | 0.011 | 0.067 | 0.004 | 0.006 | 0.554 | 0.144 | Low coordination | Low development |
| 2000 | 0.071 | 0.009 | 0.010 | 0.059 | 0.004 | 0.010 | 0.587 | 0.147 | Low coordination | Low development |
| 2005 | 0.072 | 0.017 | 0.012 | 0.063 | 0.004 | 0.012 | 0.628 | 0.157 | Low coordination | Moderate development |
| 2010 | 0.072 | 0.026 | 0.010 | 0.049 | 0.004 | 0.013 | 0.667 | 0.158 | Low coordination | Moderate development |
| 2015 | 0.077 | 0.031 | 0.011 | 0.016 | 0.004 | 0.012 | 0.660 | 0.148 | Low coordination | Moderate development |
| 2016 | 0.077 | 0.039 | 0.012 | 0.015 | 0.004 | 0.012 | 0.638 | 0.148 | Low coordination | Moderate development |
| 2017 | 0.076 | 0.036 | 0.013 | 0.013 | 0.003 | 0.012 | 0.644 | 0.146 | Low coordination | Moderate development |
| 2018 | 0.076 | 0.034 | 0.014 | 0.012 | 0.004 | 0.013 | 0.663 | 0.149 | Low coordination | Moderate grinding |
| 2019 | 0.076 | 0.034 | 0.016 | 0.011 | 0.004 | 0.013 | 0.661 | 0.148 | Low coordination | Moderate development |

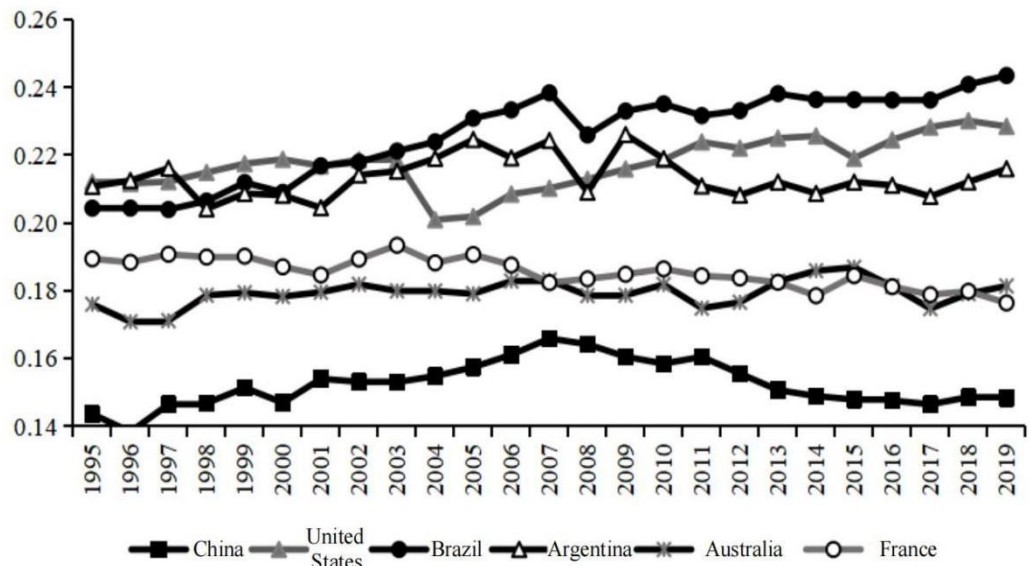

**Figure 2.** Comparison of the coupling coordination degree of the subsystems of the beef cattle industry in different countries from 1995 to 2019.

*4.5. Analysis of the Relative Development Index of the Subsystem Competitiveness*

From the relative development degrees of the two subsystems (Table 5), the quality subsystem of China's beef cattle industry significantly lags behind the development of the consumption (U3/U4 = 9.041), environment (U6/U4 = 7.687), production (U2/U4 = 5.293), resource (U4/U1 = 0.059) and trade (U4/U5 = 0.137) subsystems, indicating that the current Chinese beef product quality development level is low. The production (U2/U1 = 0.305), consumption (U3/U1 = 0.525), quality (U4/U1 = 0.059), trade (U5/U1 = 0.640) and environment (U6/U1 = 0.442) systems all lag behind the resource endowment system, indicating that these five subsystems have not yet been able to fully utilize the resource endowment to achieve the feasible relative level of development. The production subsystem lags behind the development level of the consumption and environmental systems. Overall, the development level of the production and quality of China's beef cattle industry significantly lags behind the development level of the other four subsystems, i.e., within the Chinese beef

cattle industry system, both production and quality systems lag behind, proving that the H3 was valid.

**Table 5.** Relative development degree of China's beef cattle industry subsystems.

| Subsystem | Relative Development Degree |
|---|---|
| Consumption/Quality (U3/U4) | 9.041 |
| Environment/Quality(U6/U4) | 7.687 |
| Production/Quality (U2/U4) | 5.293 |
| Trade/Environment (U5/U6) | 1.816 |
| Consumption/Environment (U3/U6) | 1.336 |
| Consumption/Trade (U3/U5) | 1.262 |
| Production/Trade (U2/U5) | 0.927 |
| Production/Environment (U2/U6) | 0.676 |
| Trade/Resources (U5/U1) | 0.640 |
| Production/Consumption (U2/U3) | 0.586 |
| Consumption/Resources (U3/U1) | 0.525 |
| Environment/Resources (U6/U1) | 0.442 |
| Production/Resources (U2/U1) | 0.305 |
| Quality/Trade (U4/U5) | 0.137 |
| Quality/Resources (U4/U1) | 0.059 |

## 5. Discussion

### 5.1. Similarities and Differences with the Existing Research

The academic research on the development level of the beef cattle industry mainly focuses on the measurement of the production efficiency and the total factor productivity. From 1980 to 2011, China's beef production increased significantly, showing a fluctuating upward trend, and the spatial transfer changed from "pastoral area" to "pastoral agricultural area". The beef-dominant areas in the northeast, northwest, southwest and Huang Huai Hai Plain gradually emerged [27]. From 1998 to 2014, the technical efficiency of beef cattle breeding in China increased year by year, with an average of 0.814. The breeding density, agricultural mechanization and epidemic risk are all factors found to affect the technical efficiency of beef cattle breeding [28]. From 2013 to 2017, the average total factor productivity of the beef cattle industry in 15 provinces and regions in China was 1.015, with an average annual growth of 1.50%. Transportation conditions and greenhouse gas emissions have an impact on the changes in the total factor productivity of the beef cattle industry [29–31]. Chinese rural residents' beef consumption is more sensitive to price and income levels than urban residents' beef consumption [32].

Many scholars have summarized the problems existing in the development of China's beef cattle industry. The current cost of beef cattle breeding in China is rising rapidly, reducing farmers' motivation. The number of beef cattle stocks and basic cows has decreased year by year, the source of cattle is in short supply, and the growth of beef production is slow. The serious dislocation between production and market conditions has led to the prominent contradiction between supply and demand [25,33–36]. There are also various problems, such as an imperfect market supervision mechanism and insufficient policy support [37,38]. The United States is a major beef cattle producer. Its typical practices and successful experiences in the application of production technology, construction of specialized and socialized service systems, cost-saving and efficiency improvement, and environmental protection are worthy of reference for China [39,40]. Compared with other countries (regions) with a better beef industry development, China lacks comparative advantages. Although China is a large beef production and consumption country, it is a small export country with a weak international competitiveness [41,42].

Previous studies on the competitiveness of animal husbandry focused on the traditional trade perspective. At present, there is no research on the high-quality development of animal husbandry from the perspective of high quality, high efficiency, safety and environmental protection. Starting from the industrial system, this study establishes

a competitiveness evaluation index system that includes six aspects: resource endowment, production, quality, consumption, trade and environment. This content enriches the previous evaluation of the competitiveness of the livestock industry, makes a more comprehensive, objective and specific evaluation of the industrial competitiveness, and provides a more comprehensive value reference and optimization policy recommendations for further improving the international competitiveness of China's beef industry.

*5.2. Limitation and Future Research Direction*

This study analyzed the development level of China's beef cattle industry from different perspectives at home and abroad. Although this study strives to be objective and rigorous in the analysis process, it still has various limitations. Due to the lack of relevant data on different varieties of beef cattle in the world, this study ignored the differences of beef cattle varieties in constructing the evaluation index of the beef cattle industry development level. The data used in this study are partially missing. Although the missing data are estimated by the relevant methods to ensure the accuracy of the data as much as possible, there will still be various inevitable errors.

## 6. Conclusions and Recommendations

*6.1. Conclusions*

This study comprehensively assesses the development level of China's beef cattle industry from the development of six industrial subsystems, resource endowment, production, consumption, quality, trade and environment, and makes international comparisons to explore the main lagging points in the development of the beef cattle industry, with the following main conclusions.

First, the development level of China's beef cattle industry is following a slowly fluctuating trend. Prior to 2008, the overall development level index of China's beef cattle industry fluctuated from 3.72 to 4.14 due to the development level of the resource endowment and production systems; after 2008, it showed a fluctuating downward trend, mainly due to the obvious decline in the development level of the trade system. There is a certain degree of variation in the changes in the development level of each subsystem. The development level of the resource endowment and the consumption subsystem has slightly increased; the development of the production subsystem is weaker but has significantly increased; the development level of the quality subsystem is more stable overall; the development level of the trade subsystem has decreased most significantly; and the development level of the environmental system has increased.

Second, the development level of China's beef cattle industry is lower than that of Brazil, Australia, Argentina and the United States. Brazil has the highest level of beef cattle industry development (mean value of 8.34), followed by Australia (mean value of 7.67), Argentina (mean value of 7.56) and the United States (mean value of 7.27), and China (mean value of 3.68) is in a low position, ranking eighth among the 12 representative countries. Compared with those of other countries, the production, consumption, quality, trade, and environmental subsystems of China's beef cattle industry are underdeveloped, except for the resource endowment.

Third, the subsystems of China's beef cattle industry are in a state of low coordination. From 1995 to 2019, the coupling degree of each subsystem showed a fluctuating change of rising-declining-rising and a trend of advancing from the low-development to the medium-development stage. Each subsystem was in a state of low coordination; i.e., the development rates between each subsystem were inconsistent. The development of the production, consumption, trade, quality and environment systems of China's beef cattle industry lags behind the development of the resource endowment system, indicating that these five subsystems have not been able to fully utilize the resource endowment to achieve relative development. Within China's current beef cattle industry system, the production and quality development are both lagging behind.

*6.2. Policy Recommendations*

To address the current situation and major shortcomings of China's beef cattle industry, this study suggests that policy guidance should be strengthened to improve production efficiency. The Chinese government can increase the motivation of female cattle breeding by resuming the "Female Herd Expansion and Enrichment Program". Training in beef cattle production skills should be strengthened to enhance the reproductive efficiency of females and the output efficiency of beef cattle. China should also learn from the experience of other countries and implement a "grass-fed" cattle production model to reduce feeding costs. Green farming and animal welfare should also be supported to promote healthy farming, improve animal welfare, and improve the overall quality of livestock products. Healthy livestock and poultry breeding is a systemic project that requires a series of technical measures to improve all aspects of livestock housing, feed mix, disease prevention and control, and resource utilization of manure to achieve the ultimate goal of healthy livestock and poultry, safe livestock products and a friendly environment. Developed countries in Europe and the United States have formed a more complete system for farm animal welfare, while China has started late and is still in the stages of the trial and implementation. It is necessary to establish and improve a series of policy guarantees, supervision and incentives through phased and the step-by-step realization to enhance the overall quality of livestock products and enhance international market recognition and competitiveness under the guarantee of environmental friendliness.

**Author Contributions:** Conceptualization, M.W., X.L. and X.M.; methodology, X.L. and X.M.; software, X.L., X.M. and H.Z.; validation, X.L., X.M. and M.W.; formal analysis, X.L., X.M. and H.Z.; investigation, X.L., X.M., H.Z. and M.W.; resources, X.L. and M.W.; data curation, X.L., X.M., M.W. and H.Z.; writing-original draft preparation, X.L. and X.M.; writing, review and editing, X.L. and M.W.; visualization, X.L. and H.Z.; supervision, M.W.; project administration, M.W.; funding acquisition, M.W. All authors have read and agreed to the published version of the manuscript.

**Funding:** We gratefully acknowledge the funding support from the Key projects of the National Natural Science Foundation of China (72033009) and the Scientific and Technological Innovation Project of the Chinese Academy of Agricultural Sciences (ASTIP-IAED-2022-01).

**Institutional Review Board Statement:** Not applicable.

**Informed Consent Statement:** Informed consent was obtained from all subjects involved in the study.

**Data Availability Statement:** The data presented in this study are available within the article.

**Conflicts of Interest:** The authors declare no conflict of interest.

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
