# Peer review of "International Evaluation of China’s Beef Cattle Industry Development Level and Lagging Points"

_agriculture, doi:10.3390/agriculture12101597_

Round 1

Reviewer 1 Report

Dear Authors and Editor,

I encourage the continuity of the paper so that it can be published, congratulations on the relevant topic!

Improvement suggestions:

- Corresponding author's email is with another font for the rest of the text.

Abstract

- I believe it would be better to exclude the word "high" in the following sentence, to remove the value judgment. “High quality, high efficiency, safety and environmental protection are important directions for the development of animal husbandry in China.”

- "The beef cattle industry in China is less developed than that in Brazil, the United States, Argentina, Australia and other countries." – In what aspects? Because of this China has import level from these countries?

- The reasons would be associated with what Ph.D. Marcos Jank writes in the link below, recognized professor in global agribusiness? 

Available in Portuguese: https://www.linkedin.com/pulse/o-casamento-inevit%C3%A1vel-entre-brasil-e-china-agroneg%C3%B3cio-marcos-jank/

Introduction

- “From China's reform and opening up to the end of the last century,” -  Important to better indicate the timeline.

- "How does the competitiveness of the beef cattle industry compare with that 84” in other countries (regions)?" - Which regions to be able to make the comparison? The reason for choosing these countries/regions. It is not clear on the text.

Related Research Status

- I suggest replacing the title, to something that can understand the theoretical framework of the research. Section can be improved with more applied studies and theoretical background.

Material in methods, perhaps: Material and Methods?

- Item 3.1 is based on a low number of references. Improve this point, which should also be attended to for other subsections on the same way.

- For Table 1 and the sentence: "Finally, the specific indicators selected to evaluate the development level of China's 194 beef cattle industry are shown in Table 1.” - The bases for consultation, citations, etc., are not clear.

- It is important to highlight the choice of time period, justifying with greater emphasis that it was from 1995 to 2019.

Results and Discussion

- In addition to being presented, the results need to be discussed in the light of previous and related studies.

- In Table 3, it is necessary to adjust the alignment, it is outside the margins.

Author Response

Dear Editors and Reviewers: We feel great thanks for your professional review work on our article. Please see the attachment for the modifications and explanations we made according to your suggestions

Reviewer 2 Report

The topic of the paper is interesting.

The main suggestions for changes are as follows:

Please reword the following sentence (L42-L44): „China's annual 42 beef production increased from 230,000 tons to 5,131,000 tons, cattle stock and slaughter 43 increased from 7,134.6 and 2,968,000 to 12,353.2 and 3,809,000, up 73.1% and 11.8 times, 44 respectively”, for a better understanding among the readers, it is not distinguished which are the increases and for which category.

Please highlight the objective of the paper at the end of the introduction section.

Please insert the hypothesis/hypotheses of the research that will be later confirmed or denied, for the research questions.

Please reposition table 3 so that all the content fits within the page limits.

Please add the limits of the research, either at the end of the conclusions or in the dedicated section.

Author Response

(The authors gave the same response as above.)
